# DiffWave: A Versatile Diffusion Model for Audio Synthesis

**Zhifeng Kong** *
Computer Science and Engineering, UCSD
z4kong@eng.ucsd.edu

**Wei Ping**
NVIDIA
wping@nvidia.com

**Jiaji Huang, Kexin Zhao**
Baidu Research
{huangjiaji,kexinzhao}@baidu.com

**Bryan Catanzaro**
NVIDIA
bcatanzaro@nvidia.com

## Abstract

In this work, we propose DiffWave, a versatile diffusion probabilistic model for conditional and unconditional waveform generation. The model is non-autoregressive, and converts the white noise signal into structured waveform through a Markov chain with a constant number of steps at synthesis. It is efficiently trained by optimizing a variant of variational bound on the data likelihood. DiffWave produces high-fidelity audio in different waveform generation tasks, including neural vocoding conditioned on mel spectrogram, class-conditional generation, and unconditional generation. We demonstrate that DiffWave matches a strong WaveNet vocoder in terms of speech quality (MOS: 4.44 versus 4.43), while synthesizing orders of magnitude faster. In particular, it significantly outperforms autoregressive and GAN-based waveform models in the challenging unconditional generation task in terms of audio quality and sample diversity from various automatic and human evaluations. [1]

## 1 Introduction

Deep generative models have produced high-fidelity raw audio in speech synthesis and music generation. In previous work, likelihood-based models, including autoregressive models (van den Oord et al., 2016; Kalchbrenner et al., 2018; Mehri et al., 2017) and flow-based models (Prenger et al., 2019; Ping et al., 2020; Kim et al., 2019), have predominated in audio synthesis because of the simple training objective and superior ability of modeling the fine details of waveform in real data. There are other waveform models, which often require auxiliary losses for training, such as flow-based models trained by distillation (van den Oord et al., 2018; Ping et al., 2019), variational auto-encoder (VAE) based model (Peng et al., 2020), and generative adversarial network (GAN) based models (Kumar et al., 2019; Bińkowski et al., 2020; Yamamoto et al., 2020).

Most of previous waveform models focus on audio synthesis with informative local conditioner (e.g., mel spectrogram or aligned linguistic features), with only a few exceptions for unconditional generation (Mehri et al., 2017; Donahue et al., 2019). It has been noticed that autoregressive models (e.g., WaveNet) tend to generate made-up word-like sounds (van den Oord et al., 2016), or inferior samples (Donahue et al., 2019) under unconditional settings. This is because very long sequences need to be generated (e.g., 16,000 time-steps for one second speech) without any conditional information.

*Diffusion probabilistic models* (diffusion models for brevity) are a class of promising generative models, which use a Markov chain to gradually convert a simple distribution (e.g., isotropic Gaussian) into complicated data distribution (Sohl-Dickstein et al., 2015; Goyal et al., 2017; Ho et al., 2020). Although the data likelihood is intractable, diffusion models can be efficiently trained by optimizing the variational lower bound (ELBO). Most recently, a certain parameterization has been shown successful in image synthesis (Ho et al., 2020), which is connected with denoising score matching (Song

---

*Contributed to the work during an internship at Baidu Research, USA.
[1] Audio samples are in: https://diffwave-demo.github.io/

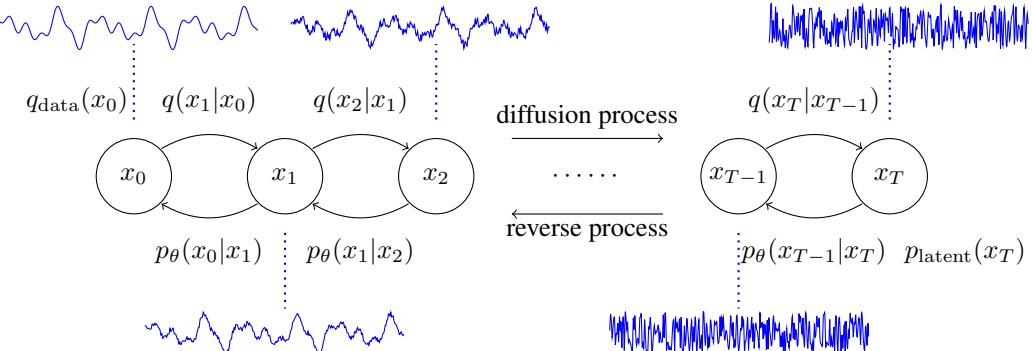

Figure 1: The diffusion and reverse process in diffusion probabilistic models. The reverse process gradually converts the white noise signal into speech waveform through a Markov chain $p_\theta(x_{t-1}|x_t)$.

& Ermon, 2019). Diffusion models can use a diffusion (noise-adding) process without learnable parameters to obtain the "whitened" latents from training data. Therefore, no additional neural networks are required for training in contrast to other models (e.g., the encoder in VAE (Kingma & Welling, 2014) or the discriminator in GAN (Goodfellow et al., 2014)). This avoids the challenging "posterior collapse" or "mode collapse" issues stemming from the joint training of two networks, and hence is valuable for high-fidelity audio synthesis.

In this work, we propose DiffWave, a versatile diffusion probabilistic model for raw audio synthesis. DiffWave has several advantages over previous work: *i*) It is *non-autoregressive* thus can synthesize high-dimensional waveform in parallel. *ii*) It is *flexible* as it does not impose any architectural constraints in contrast to flow-based models, which need to keep the bijection between latents and data (e.g., see more analysis in Ping et al. (2020)). This leads to small-footprint neural vocoders that still generate high-fidelity speech. *iii*) It uses a *single* ELBO-based training objective without any auxiliary losses (e.g., spectrogram-based losses) for high-fidelity synthesis. *iv*) It is a *versatile* model that produces high-quality audio signals for both conditional and unconditional waveform generation.

Specifically, we make the following contributions:

1. DiffWave uses a feed-forward and *bidirectional* dilated convolution architecture motivated by WaveNet (van den Oord et al., 2016). It matches the strong WaveNet vocoder in terms of speech quality (MOS: $4.44$ vs. $4.43$), while synthesizing orders of magnitude faster as it only requires a few sequential steps (e.g., 6) for generating very long waveforms.

2. Our small DiffWave has 2.64M parameters and synthesizes 22.05 kHz high-fidelity speech (MOS: $4.37$) more than $5\times$ faster than real-time on a V100 GPU without engineered kernels. Although it is still slower than the state-of-the-art flow-based models (Ping et al., 2020; Prenger et al., 2019), it has much smaller footprint. We expect further speed-up by optimizing its inference mechanism in the future.

3. DiffWave significantly outperforms WaveGAN (Donahue et al., 2019) and WaveNet in the challenging unconditional and class-conditional waveform generation tasks in terms of audio quality and sample diversity measured by several automatic and human evaluations.

We organize the rest of the paper as follows. We present the diffusion models in Section 2, and introduce DiffWave architecture in Section 3. Section 4 discusses related work. We report experimental results in Section 5 and conclude the paper in Section 6.

## 2 DIFFUSION PROBABILISTIC MODELS

We define $q_{\text{data}}(x_0)$ as the data distribution on $\mathbb{R}^L$, where $L$ is the data dimension. Let $x_t \in \mathbb{R}^L$ for $t = 0, 1, \cdots, T$ be a sequence of variables with the same dimension, where $t$ is the index for diffusion steps. Then, a diffusion model of $T$ steps is composed of two processes: the diffusion process, and the reverse process (Sohl-Dickstein et al., 2015). Both of them are illustrated in Figure 1.

| **Algorithm 1** Training | **Algorithm 2** Sampling |
|---|---|
| **for** $i = 1, 2, \cdots, N_{\text{iter}}$ **do**
    Sample $x_0 \sim q_{\text{data}}$, $\epsilon \sim \mathcal{N}(0, I)$, and
        $t \sim \text{Uniform}(\{1, \cdots, T\})$
    Take gradient step on
        $\nabla_\theta \|\epsilon - \epsilon_\theta(\sqrt{\bar{\alpha}_t}x_0 + \sqrt{1 - \bar{\alpha}_t}\epsilon,\ t)\|_2^2$
    according to Eq. (7)
**end for** | Sample $x_T \sim p_{\text{latent}} = \mathcal{N}(0, I)$
**for** $t = T, T-1, \cdots, 1$ **do**
    Compute $\mu_\theta(x_t, t)$ and $\sigma_\theta(x_t, t)$ using Eq. (5)
    Sample $x_{t-1} \sim p_\theta(x_{t-1}|x_t) =$
        $\mathcal{N}(x_{t-1}; \mu_\theta(x_t, t), \sigma_\theta(x_t, t)^2 I)$
**end for**
**return** $x_0$ |

The **diffusion process** is defined by a fixed Markov chain from data $x_0$ to the latent variable $x_T$:

$$q(x_1, \cdots, x_T | x_0) = \prod_{t=1}^{T} q(x_t | x_{t-1}), \tag{1}$$

where each of $q(x_t|x_{t-1})$ is fixed to $\mathcal{N}(x_t; \sqrt{1 - \beta_t}x_{t-1}, \beta_t I)$ for a small positive constant $\beta_t$. The function of $q(x_t|x_{t-1})$ is to add small Gaussian noise to the distribution of $x_{t-1}$. The whole process gradually converts data $x_0$ to whitened latents $x_T$ according to a variance schedule $\beta_1, \cdots, \beta_T$. [2]

The **reverse process** is defined by a Markov chain from $x_T$ to $x_0$ parameterized by $\theta$:

$$p_{\text{latent}}(x_T) = \mathcal{N}(0, I), \text{ and } p_\theta(x_0, \cdots, x_{T-1}|x_T) = \prod_{t=1}^{T} p_\theta(x_{t-1}|x_t), \tag{2}$$

where $p_{\text{latent}}(x_T)$ is isotropic Gaussian, and the transition probability $p_\theta(x_{t-1}|x_t)$ is parameterized as $\mathcal{N}(x_{t-1}; \mu_\theta(x_t, t), \sigma_\theta(x_t, t)^2 I)$ with shared parameter $\theta$. Note that both $\mu_\theta$ and $\sigma_\theta$ take two inputs: the diffusion-step $t \in \mathbb{N}$, and variable $x_t \in \mathbb{R}^L$. $\mu_\theta$ outputs an $L$-dimensional vector as the mean, and $\sigma_\theta$ outputs a real number as the standard deviation. The goal of $p_\theta(x_{t-1}|x_t)$ is to eliminate the Gaussian noise (i.e. denoise) added in the diffusion process.

**Sampling**: Given the reverse process, the generative procedure is to first sample an $x_T \sim \mathcal{N}(0, I)$, and then sample $x_{t-1} \sim p_\theta(x_{t-1}|x_t)$ for $t = T, T-1, \cdots, 1$. The output $x_0$ is the sampled data.

**Training**: The likelihood $p_\theta(x_0) = \int p_\theta(x_0, \cdots, x_{T-1}|x_T) \cdot p_{\text{latent}}(x_T) \, \mathrm{d}x_{1:T}$ is intractable to calculate in general. The model is thus trained by maximizing its variational lower bound (ELBO):

$$\mathbb{E}_{q_{\text{data}}(x_0)} \log p_\theta(x_0) = \mathbb{E}_{q_{\text{data}}(x_0)} \log \mathbb{E}_{q(x_1, \cdots, x_T | x_0)} \left[ \frac{p_\theta(x_0, \cdots, x_{T-1}|x_T) \times p_{\text{latent}}(x_T)}{q(x_1, \cdots, x_T | x_0)} \right]$$

$$\geq \mathbb{E}_{q(x_0, \cdots, x_T)} \log \frac{p_\theta(x_0, \cdots, x_{T-1}|x_T) \times p_{\text{latent}}(x_T)}{q(x_1, \cdots, x_T | x_0)} := \text{ELBO}. \tag{3}$$

Most recently, Ho et al. (2020) showed that under a certain parameterization, the ELBO of the diffusion model can be calculated in closed-form. This accelerates the computation and avoids Monte Carlo estimates, which have high variance. This parameterization is motivated by its connection to denoising score matching with Langevin dynamics (Song & Ermon, 2019; 2020). To introduce this parameterization, we first define some constants based on the variance schedule $\{\beta_t\}_{t=1}^T$ in the diffusion process as in Ho et al. (2020):

$$\alpha_t = 1 - \beta_t, \quad \bar{\alpha}_t = \prod_{s=1}^{t} \alpha_s, \quad \tilde{\beta}_t = \frac{1 - \bar{\alpha}_{t-1}}{1 - \bar{\alpha}_t}\beta_t \quad \text{for } t > 1 \text{ and } \tilde{\beta}_1 = \beta_1. \tag{4}$$

Then, the parameterizations of $\mu_\theta$ and $\sigma_\theta$ are defined by

$$\mu_\theta(x_t, t) = \frac{1}{\sqrt{\alpha_t}} \left( x_t - \frac{\beta_t}{\sqrt{1 - \bar{\alpha}_t}} \epsilon_\theta(x_t, t) \right), \text{ and } \sigma_\theta(x_t, t) = \tilde{\beta}_t^{\frac{1}{2}}, \tag{5}$$

where $\epsilon_\theta : \mathbb{R}^L \times \mathbb{N} \to \mathbb{R}^L$ is a neural network also taking $x_t$ and the diffusion-step $t$ as inputs. Note that $\sigma_\theta(x_t, t)$ is fixed to a constant $\tilde{\beta}_t^{\frac{1}{2}}$ for every step $t$ under this parameterization. In the following proposition, we explicitly provide the closed-form expression of the ELBO.

---

[2]One can find that $q(x_T|x_0)$ approaches to isotropic Gaussian with large $T$ in Eq. (11) in the Appendix A.

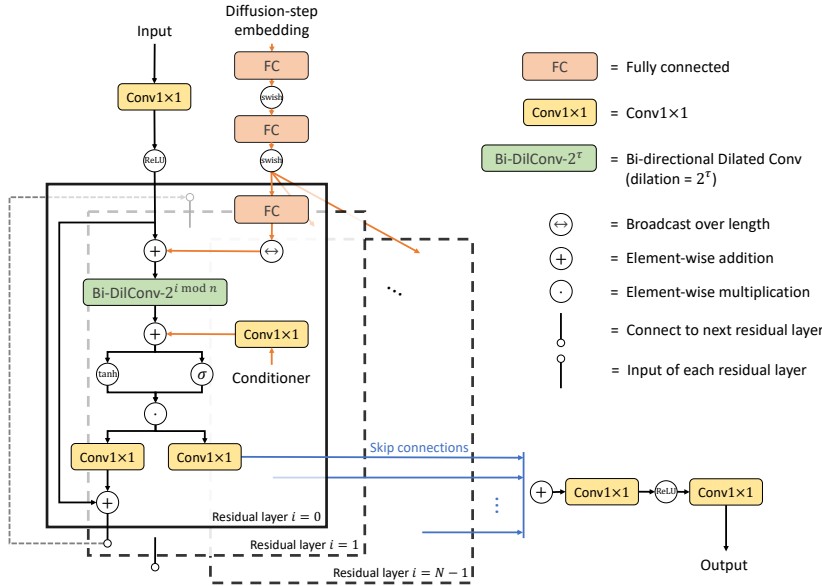

Figure 2: The network architecture of DiffWave in modeling $\epsilon_\theta : \mathbb{R}^L \times \mathbb{N} \to \mathbb{R}^L$.

**Proposition 1.** *(Ho et al., 2020) Suppose a series of fixed schedule $\{\beta_t\}_{t=1}^T$ are given. Let $\epsilon \sim \mathcal{N}(0, I)$ and $x_0 \sim q_{\text{data}}$. Then, under the parameterization in Eq. (5), we have*

$$-\text{ELBO} = c + \sum_{t=1}^{T} \kappa_t \mathbb{E}_{x_0, \epsilon} \left\| \epsilon - \epsilon_\theta(\sqrt{\bar{\alpha}_t} x_0 + \sqrt{1 - \bar{\alpha}_t}\epsilon, \ t) \right\|_2^2 \tag{6}$$

*for some constants $c$ and $\kappa_t$, where $\kappa_t = \frac{\beta_t}{2\alpha_t(1 - \bar{\alpha}_{t-1})}$ for $t > 1$, and $\kappa_1 = \frac{1}{2\alpha_1}$.*

Note that $c$ is irrelevant for optimization purpose. The key idea in the proof is to expand the ELBO into a sum of KL divergences between tractable Gaussian distributions, which have a closed-form expression. We refer the readers to look at Section A in the Appendix for the full proof.

In addition, Ho et al. (2020) reported that minimizing the following unweighted variant of the ELBO leads to higher generation quality:

$$\min_\theta L_{\text{unweighted}}(\theta) = \mathbb{E}_{x_0, \epsilon, t} \left\| \epsilon - \epsilon_\theta(\sqrt{\bar{\alpha}_t} x_0 + \sqrt{1 - \bar{\alpha}_t}\epsilon, \ t) \right\|_2^2 \tag{7}$$

where $t$ is uniformly taken from $1, \cdots, T$. Therefore, we also use this training objective in this paper. We summarize the training and sampling procedures in Algorithm 1 and 2, respectively.

**Fast sampling**: Given a trained model from Algorithm 1, we noticed that the most effective denoising steps at sampling occur near $t = 0$ (see Section IV on demo website). This encourages us to design a fast sampling algorithm with much fewer denoising steps $T_{\text{infer}}$ (e.g., 6) than $T$ at training (e.g., 200). The key idea is to "collapse" the $T$-step reverse process into a $T_{\text{infer}}$-step process with carefully designed variance schedule. We provide the details in Appendix B.

## 3 DIFFWAVE ARCHITECTURE

In this section, we present the architecture of DiffWave (see Figure 2 for an illustration). We build the network $\epsilon_\theta : \mathbb{R}^L \times \mathbb{N} \to \mathbb{R}^L$ in Eq. (5) based on a *bidirectional* dilated convolution architecture that is different from WaveNet (van den Oord et al., 2016), because there is no autoregressive generation constraint. [3] The similar architecture has been applied for source separation (Rethage et al., 2018; Lluís et al., 2018). The network is non-autoregressive, so generating an audio $x_0$ with length $L$ from latents $x_T$ requires $T$ rounds of forward propagation, where $T$ (e.g., 50) is much smaller than the waveform length $L$. The network is composed of a stack of $N$ residual layers with residual channels

---

[3]Indeed, we found the causal dilated convolution architecture leads to worse audio quality in DiffWave.

$C$. These layers are grouped into $m$ blocks and each block has $n = \frac{N}{m}$ layers. We use a bidirectional dilated convolution (Bi-DilConv) with kernel size 3 in each layer. The dilation is doubled at each layer within each block, i.e., $[1, 2, 4, \cdots, 2^{n-1}]$. We sum the skip connections from all residual layers as in WaveNet. More details including the tensor shapes are included in Section C in the Appendix.

### 3.1 Diffusion-step Embedding

It is important to include the diffusion-step $t$ as part of the input, as the model needs to output different $\epsilon_\theta(\cdot, t)$ for different $t$. We use an 128-dimensional encoding vector for each $t$ (Vaswani et al., 2017):

$$t_{\text{embedding}} = \left[\sin\left(10^{\frac{0\times 4}{63}} t\right), \cdots, \sin\left(10^{\frac{63\times 4}{63}} t\right), \cos\left(10^{\frac{0\times 4}{63}} t\right), \cdots, \cos\left(10^{\frac{63\times 4}{63}} t\right)\right] \quad (8)$$

We then apply three fully connected (FC) layers on the encoding, where the first two FCs share parameters among all residual layers. The last residual-layer-specific FC maps the output of the second FC into a $C$-dimensional embedding vector. We next broadcast this embedding vector over length and add it to the input of every residual layer.

### 3.2 Conditional generation

**Local conditioner:** In speech synthesis, a neural vocoder can synthesize the waveform conditioned on the aligned linguistic features (van den Oord et al., 2016; Arık et al., 2017b), the mel spectrogram from a text-to-spectrogram model (Ping et al., 2018; Shen et al., 2018), or the hidden states within the text-to-wave architecture (Ping et al., 2019; Donahue et al., 2020). In this work, we test DiffWave as a neural vocoder conditioned on mel spectrogram. We first upsample the mel spectrogram to the same length as waveform through transposed 2-D convolutions. After a layer-specific Conv1×1 mapping its mel-band into $2C$ channels, the conditioner is added as a bias term for the dilated convolution in each residual layer. The hyperparameters can be found in Section 5.1.

**Global conditioner:** In many generative tasks, the conditional information is given by global discrete labels (e.g., speaker IDs or word IDs). We use shared embeddings with dimension $d_{\text{label}} = 128$ in all experiments. In each residual layer, we apply a layer-specific Conv1×1 to map $d_{\text{label}}$ to $2C$ channels, and add the embedding as a bias term after the dilated convolution in each residual layer.

### 3.3 Unconditional generation

In unconditional generation task, the model needs to generate consistent utterances without conditional information. It is important for the output units of the network to have a receptive field size (denoted as $r$) larger than the length $L$ of the utterance. Indeed, we need $r \geq 2L$, thus the left and right-most output units have receptive fields covering the whole $L$-dimensional inputs as illustrated in Figure 4 in Appendix. This posts a challenge for architecture design even with the dilated convolutions.

For a stack of dilated convolution layers, the receptive field size of the output is up to: $r = (k-1)\sum_i d_i + 1$, where $k$ is the kernel size and $d_i$ is the dilation at $i$-th residual layer. For example, 30-layer dilated convolution has a receptive field size $r = 6139$, with $k = 3$ and dilation cycle $[1, 2, \cdots, 512]$. This only amounts to 0.38s of 16kHz audio. We can further increase the number of layers and the size of dilation cycles; however, we found degraded quality with deeper layers and larger dilation cycles. This is particularly true for WaveNet. In fact, previous study (Shen et al., 2018) suggests that even a moderate large receptive field size (e.g., 6139) is not effectively used in WaveNet and it tends to focus on much shorter context (e.g., 500). DiffWave has an advantage in enlarging the receptive fields of output $x_0$: by iterating from $x_T$ to $x_0$ in the reverse process, the receptive field size can be increased up to $T \times r$, which makes DiffWave suitable for unconditional generation.

## 4 Related Work

In the past years, many neural text-to-speech (TTS) systems have been introduced. An incomplete list includes WaveNet (van den Oord et al., 2016), Deep Voice 1 & 2 & 3 (Arık et al., 2017a;b; Ping et al., 2018), Tacotron 1 & 2 (Wang et al., 2017; Shen et al., 2018), Char2Wav (Sotelo et al., 2017), VoiceLoop (Taigman et al., 2018), Parallel WaveNet (van den Oord et al., 2018), WaveRNN (Kalchbrenner et al., 2018), ClariNet (Ping et al., 2019), ParaNet (Peng et al., 2020), FastSpeech (Ren et al., 2019), GAN-TTS (Bińkowski et al., 2020), and Flowtron (Valle et al., 2020). These systems first

generate intermediate representations (e.g., aligned linguistic features, mel spectrogram, or hidden representations) conditioned on text, then use a neural vocoder to synthesize the raw waveform.

Neural vocoder plays the most important role in the recent success of speech synthesis. Autoregressive models like WaveNet and WaveRNN can generate high-fidelity speech, but in a sequential way of generation. Parallel WaveNet and ClariNet distill parallel flow-based models from WaveNet, thus can synthesize waveform in parallel. In contrast, WaveFlow (Ping et al., 2020), WaveGlow (Prenger et al., 2019) and FloWaveNet (Kim et al., 2019) are trained by maximizing likelihood. There are other waveform models, such as VAE-based models (Peng et al., 2020), GAN-based models (Kumar et al., 2019; Yamamoto et al., 2020; Bińkowski et al., 2020), and neural signal processing models (Wang et al., 2019; Engel et al., 2020; Ai & Ling, 2020). In contrast to likelihood-based models, they often require auxiliary training losses to improve the audio fidelity. The proposed DiffWave is another promising neural vocoder synthesizing the best quality of speech with a single objective function.

Unconditional generation of audio in the time domain is a challenging task in general. Likelihood-based models are forced to learn all possible variations within the dataset without any conditional information, which can be quite difficult with limited model capacity. In practice, these models produce made-up word-like sounds or inferior samples (van den Oord et al., 2016; Donahue et al., 2019). VQ-VAE (van den Oord et al., 2017) circumvents this issue by compressing the waveform into compact latent code, and training an autoregressive model in latent domain. GAN-based models are believed to be suitable for unconditional generation (e.g., Donahue et al., 2019) due to the "mode seeking" behaviour and success in image domain (Brock et al., 2018). Note that unconditional generation of audio in the frequency domain is considered easier, as the spectrogram is much shorter (e.g., 200×) than waveform (Vasquez & Lewis, 2019; Engel et al., 2019; Palkama et al., 2020).

In this work, we demonstrate the superior performance of DiffWave in unconditional generation of waveform. In contrast to the exact-likelihood models, DiffWave maximizes a variational lower bound of the likelihood, which can focus on the major variations within the data and alleviate the requirements for model capacity. In contrast to GAN or VAE-based models (Donahue et al., 2019; Peng et al., 2020), it is much easier to train without mode collapse, posterior collapse, or training instability stemming from the joint training of two networks. There is a concurrent work (Chen et al., 2020) that uses diffusion probabilistic models for waveform generation. In contrast to DiffWave, it uses a neural architecture similar to GAN-TTS and focuses on the neural vocoding task only. Our DiffWave vocoder has much fewer parameters than WaveGrad – 2.64M vs. 15M for Base models and 6.91M vs. 23M for Large models. The small memory footprint is preferred in production TTS systems, especially for on-device deployment. In addition, DiffWave requires a smaller batch size (16 vs. 256) and fewer computational resources for training.

## 5 EXPERIMENTS

We evaluate DiffWave on neural vocoding, unconditional and class-conditional generation tasks.

### 5.1 NEURAL VOCODING

**Data:** We use the LJ speech dataset (Ito, 2017) that contains ∼24 hours of audio recorded in home environment with a sampling rate of 22.05 kHz. It contains 13,100 utterances from a female speaker.

**Models:** We compare DiffWave with several state-of-the-art neural vocoders, including WaveNet, ClariNet, WaveGlow and WaveFlow. Details of baseline models can be found in the original papers. Their hyperparameters can be found in Table 1. Our DiffWave models have 30 residual layers, kernel size 3, and dilation cycle $[1, 2, \cdots, 512]$. We compare DiffWave models with different number of diffusion steps $T \in \{20, 40, 50, 200\}$ and residual channels $C \in \{64, 128\}$. We use linear spaced schedule for $\beta_t \in [1 \times 10^{-4}, 0.02]$ for DiffWave with $T = 200$, and $\beta_t \in [1 \times 10^{-4}, 0.05]$ for DiffWave with $T \leq 50$. The reason to increase $\beta_t$ for smaller $T$ is to make $q(x_T|x_0)$ close to $p_{\text{latent}}$. In addition, we compare the fast sampling algorithm with smaller $T_{\text{infer}}$ (see Appendix B), denoted as DiffWave (Fast), with the regular sampling (Algorithm 2). Both of them use the same trained models.

**Conditioner:** We use the 80-band mel spectrogram of the original audio as the conditioner to test these neural vocoders as in previous work (Ping et al., 2019; Prenger et al., 2019; Kim et al., 2019). We set FFT size to 1024, hop size to 256, and window size to 1024. We upsample the mel spectrogram 256 times by applying two layers of transposed 2-D convolution (in time and frequency) interleaved

Table 1: The model hyperparameters, model footprint, and 5-scale Mean Opinion Score (MOS) with 95% confidence intervals for WaveNet, ClariNet, WaveFlow, WaveGlow and the proposed DiffWave on the **neural vocoding** task. ↑ means the number is the higher the better, and ↓ means the number is the lower the better.

| Model | $T$ | $T_{\text{infer}}$ | layers | res. channels | #param($\downarrow$) | MOS($\uparrow$) |
|---|---|---|---|---|---|---|
| WaveNet | — | — | 30 | 128 | 4.57M | **4.43** $\pm 0.10$ |
| ClariNet | — | — | 60 | 64 | 2.17M | $4.27 \pm 0.09$ |
| WaveGlow | — | — | 96 | 256 | 87.88M | $4.33 \pm 0.12$ |
| WaveFlow | — | — | 64 | 64 | 5.91M | $4.30 \pm 0.11$ |
| WaveFlow | — | — | 64 | 128 | 22.25M | $4.40 \pm 0.07$ |
| DiffWave $_{\text{BASE}}$ | 20 | 20 | 30 | 64 | 2.64M | $4.31 \pm 0.09$ |
| DiffWave $_{\text{BASE}}$ | 40 | 40 | 30 | 64 | 2.64M | $4.35 \pm 0.10$ |
| DiffWave $_{\text{BASE}}$ | 50 | 50 | 30 | 64 | 2.64M | **4.38** $\pm 0.08$ |
| DiffWave $_{\text{LARGE}}$ | 200 | 200 | 30 | 128 | 6.91M | **4.44** $\pm 0.07$ |
| DiffWave $_{\text{BASE}}$ (Fast) | 50 | 6 | 30 | 64 | 2.64M | *4.37* $\pm 0.07$ |
| DiffWave $_{\text{LARGE}}$ (Fast) | 200 | 6 | 30 | 128 | 6.91M | *4.42* $\pm 0.09$ |
| Ground-truth | — | — | — | — | — | $4.52 \pm 0.06$ |

with leaky ReLU ($\alpha = 0.4$). For each layer, the upsamling stride in time is 16 and 2-D filter sizes are $[32, 3]$. After upsampling, we use a layer-specific Conv1×1 to map the 80 mel bands into 2× residual channels, then add the conditioner as a bias term for the dilated convolution before the gated-tanh nonlinearities in each residual layer.

**Training:** We train DiffWave on 8 Nvidia 2080Ti GPUs using random short audio clips of 16,000 samples from each utterance. We use Adam optimizer (Kingma & Ba, 2015) with a batch size of 16 and learning rate $2 \times 10^{-4}$. We train all DiffWave models for 1M steps. For other models, we follow the training setups as in the original papers.

**Results:** We use the crowdMOS tookit (Ribeiro et al., 2011) for speech quality evaluation, where the test utterances from all models were presented to Mechanical Turk workers. We report the 5-scale Mean Opinion Scores (MOS), and model footprints in Table 1 [4]. Our DiffWave $_{\text{LARGE}}$ model with residual channels 128 matches the strong WaveNet vocoder in terms of speech quality (MOS: 4.44 vs. 4.43). The DiffWave $_{\text{BASE}}$ with residual channels 64 also generates high quality speech (e.g., MOS: 4.35) even with small number of diffusion steps (e.g., $T = 40$ or 20). For synthesis speed, DiffWave $_{\text{BASE}}$ ($T = 20$) in FP32 generates audio 2.1× faster than real-time, and DiffWave $_{\text{BASE}}$ ($T = 40$) in FP32 is 1.1× faster than real-time on a Nvidia V100 GPU without engineering optimization. Meanwhile, DiffWave $_{\text{BASE}}$ (Fast) and DiffWave $_{\text{LARGE}}$ (Fast) can be 5.6× and 3.5× faster than real-time respectively and still obtain good audio fidelity. In contrast, a WaveNet implementation can be 500× slower than real-time at synthesis without engineered kernels. DiffWave is still slower than the state-of-the-art flow-based models (e.g., a 5.91M WaveFlow is > 40× faster than real-time in FP16), but has smaller footprint and slightly better quality. Because DiffWave does not impose any architectural constraints as in flow-based models, we expect further speed-up by optimizing the architecture and inference mechanism in the future.

## 5.2 Unconditional generation

In this section, we apply DiffWave to an unconditional generation task based on raw waveform only.

**Data:** We use the Speech Commands dataset (Warden, 2018), which contains many spoken words by thousands of speakers under various recording conditions including some very noisy environment. We select the subset that contains spoken digits (0∼9), which we call the SC09 dataset. The SC09 dataset contains 31,158 training utterances (∼8.7 hours in total) by 2,032 speakers, where each audio has length equal to one second under sampling rate 16kHz. Therefore, the data dimension $L$ is 16,000. Note that the SC09 dataset exhibits various variations (e.g., contents, speakers, speech rate, recording conditions); the generative models need to model them without any conditional information.

**Models:** We compare DiffWave with WaveNet and WaveGAN. We also tried to remove the mel conditioner in a state-of-the-art GAN-based neural vocoder (Yamamoto et al., 2020), but found it could

---

[4] The MOS evaluation for DiffWave(Fast) with $T_{\text{infer}} = 6$ was done after paper submission and may not be directly comparable to previous scores.

Table 2: The automatic evaluation metrics (FID, IS, mIS, AM, and NDB/$K$), and 5-scale MOS with 95% confidence intervals for WaveNet, WaveGAN, and DiffWave on the **unconditional** generation task. ↑ means the number is the higher the better, and ↓ means the number is the lower the better.

| Model | FID($\downarrow$) | IS($\uparrow$) | mIS($\uparrow$) | AM($\downarrow$) | NDB/$K$($\downarrow$) | MOS($\uparrow$) |
|---|---|---|---|---|---|---|
| WaveNet-128 | 3.279 | 2.54 | 7.6 | 1.368 | 0.86 | $1.34 \pm 0.29$ |
| WaveNet-256 | 2.947 | 2.84 | 10.0 | 1.260 | 0.86 | $1.43 \pm 0.30$ |
| WaveGAN | 1.349 | 4.53 | 36.6 | 0.796 | 0.78 | $2.03 \pm 0.33$ |
| DiffWave | **1.287** | **5.30** | **59.4** | **0.636** | **0.74** | $\mathbf{3.39} \pm 0.32$ |
| Trainset | 0.000 | 8.48 | 281.4 | 0.164 | 0.00 | — |
| Testset | 0.011 | 8.47 | 275.2 | 0.166 | 0.10 | $3.72 \pm 0.28$ |

Table 3: The automatic evaluation metrics (Accuracy, FID-class, IS, mIS), and 5-scale MOS with 95% confidence intervals for WaveNet and DiffWave on the **class-conditional** generation task.

| Model | Accuracy($\uparrow$) | FID-class($\downarrow$) | IS($\uparrow$) | mIS($\uparrow$) | MOS($\uparrow$) |
|---|---|---|---|---|---|
| WaveNet-128 | 56.20% | 7.876$\pm$2.469 | 3.29 | 15.8 | $1.46 \pm 0.30$ |
| WaveNet-256 | 60.70% | 6.954$\pm$2.114 | 3.46 | 18.9 | $1.58 \pm 0.36$ |
| DiffWave | 91.20% | 1.113$\pm$0.569 | 6.63 | 117.4 | $\mathbf{3.50} \pm 0.31$ |
| DiffWave (deep & thin) | **94.00%** | **0.932**$\pm$**0.450** | **6.92** | **133.8** | $3.44 \pm 0.36$ |
| Trainset | 99.06% | 0.000$\pm$0.000 | 8.48 | 281.4 | — |
| Testset | 98.76% | 0.044$\pm$0.016 | 8.47 | 275.2 | $3.72 \pm 0.28$ |

not generate intelligible speech in this unconditional task. We use 30 layer-WaveNet models with residual channels 128 (denoted as WaveNet-128) and 256 (denoted as WaveNet-256), respectively. We tried to increase the size of the dilation cycle and the number of layers, but these modifications lead to worse quality. In particular, a large dilation cycle (e.g., up to 2048) leads to unstable training. For WaveGAN, we use their pretrained model on Google Colab. We use a 36-layer DiffWave model with kernel size 3 and dilation cycle $[1, 2, \cdots, 2048]$. We set the number of diffusion steps $T = 200$ and residual channels $C = 256$. We use linear spaced schedule for $\beta_t \in [1 \times 10^{-4}, 0.02]$.

**Training:** We train WaveNet and DiffWave on 8 Nvidia 2080Ti GPUs using full utterances. We use Adam optimizer with a batch size of 16. For WaveNet, we set the initial learning rate as $1 \times 10^{-3}$ and halve the learning rate every 200K iterations. For DiffWave, we fix the learning rate to $2 \times 10^{-4}$. We train WaveNet and DiffWave for 1M steps.

**Evaluation:** For human evaluation, we report the 5-scale MOS for speech quality similar to Section 5.1. To automatically evaluate the quality of generated audio samples, we train a ResNeXT classifier (Xie et al., 2017) on the SC09 dataset according to an open repository (Xu & Tuguldur, 2017). The classifier achieves 99.06% accuracy on the trainset and 98.76% accuracy on the testset. We use the following evaluation metrics based on the 1024-dimensional feature vector and the 10-dimensional logits from the ResNeXT classifier (see Section D in the Appendix for the detailed definitions):

- **Fréchet Inception Distance (FID)** (Heusel et al., 2017) measures both quality and diversity of generated samples, and favors generators that match moments in the feature space.

- **Inception Score (IS)** (Salimans et al., 2016) measures both quality and diversity of generated samples, and favors generated samples that can be clearly determined by the classifier.

- **Modified Inception Score (mIS)** (Gurumurthy et al., 2017) measures the within-class diversity of samples in addition to IS.

- **AM Score** (Zhou et al., 2017) takes into consideration the marginal label distribution of training data compared to IS.

- **Number of Statistically-Different Bins (NDB)** (Richardson & Weiss, 2018) measures diversity of generated samples.

**Results:** We randomly generate 1,000 audio samples from each model for evaluation. We report results in Table 2. Our DiffWave model outperforms baseline models under all metrics, including both automatic and human evaluation. Notably, the quality of audio samples generated by DiffWave is much higher than WaveNet and WaveGAN baselines (MOS: 3.39 vs. 1.43 and 2.03). Note that the quality of ground-truth audio exhibits large variations. The automatic evaluation metrics also indicate that DiffWave is better at quality, diversity, and matching marginal label distribution of training data.

### 5.3 CLASS-CONDITIONAL GENERATION

In this section, we provide the digit labels as the conditioner in DiffWave and compare our model to WaveNet. We omit the comparison with conditional WaveGAN due to its noisy output audio (Lee et al., 2018). For both DiffWave and WaveNet, the label conditioner is added to the model according to Section 3.2. We use the same dataset, model hyperparameters, and training settings as in Section 5.2.

**Evaluation:** We use slightly different automatic evaluation methods in this section because audio samples are generated according to pre-specified discrete labels. The AM score and NDB are removed because they are less meaningful when the prior label distribution of generated data is specified. We keep IS and mIS because IS favors sharp, clear samples and mIS measures within-class diversity. We modify FID to FID-class: for each *digit* from 0 to 9, we compute FID between the generated audio samples that are pre-specified as this *digit* and training utterances with the same *digit* labels, and report the mean and standard deviation of these ten FID scores. We also report classification accuracy based on the ResNeXT classifier used in Section 5.2.

**Results:** We randomly generate 100 audio samples for each digit (0 to 9) from all models for evaluation. We report results in Table 3. Our DiffWave model significantly outperforms WaveNet on all evaluation metrics. It produces superior quality than WaveNet (MOS: 3.50 vs. 1.58), and greatly decreases the gap to ground-truth (the gap between DiffWave and ground-truth is ∼10% of the gap between WaveNet and ground-truth). The automatic evaluation metrics indicate that DiffWave is much better at speech clarity ($> 91\%$ accuracy) and within-class diversity (its mIS is $6\times$ higher than WaveNet). We additionally found a deep and thin version of DiffWave with residual channels $C = 128$ and 48 residual layers can achieve slightly better accuracy but lower audio quality. One may also compare quality of generated audio samples between conditional and unconditional generation based on IS, mIS, and MOS. For both WaveNet and DiffWave, IS increases by $>20\%$, mIS almost doubles, and MOS increases by $\geq 0.11$. These results indicate that the digit labels reduces the difficulty of the generative task and helps improving the generation quality of WaveNet and DiffWave.

### 5.4 ADDITIONAL RESULTS

**Zero-shot speech denoising:** The unconditional DiffWave model can readily perform speech denoising. The SC09 dataset provides six types of noises for data augmentation in recognition tasks: white noise, pink noise, running tap, exercise bike, dude miaowing, and doing the dishes. These noises are not used during the training phase of our unconditional DiffWave in Section 5.2. We add 10% of each type of noise to test data, feed these noisy utterances into the reverse process at $t = 25$, and then obtain the outputs $x_0$'s. The audio samples are in Section V on the demo website. Note that our model is not trained on a denoising task and has zero knowledge about any noise type other than the white noise added in diffusion process. It indicates DiffWave learns a good prior of raw audio.

**Interpolation in latent space:** We can do interpolation with the digit conditioned DiffWave model in Section 5.3 on the SC09 dataset. The interpolation of voices $x_0^a, x_0^b$ between two speakers $a, b$ is done in the latent space at $t = 50$. We first sample $x_t^a \sim q(x_t|x_0^a)$ and $x_t^b \sim q(x_t|x_0^b)$ for the two speakers. We then do linear interpolation between $x_t^a$ and $x_t^b$: $x_t^\lambda = (1 - \lambda)x_t^a + \lambda x_t^b$ for $0 < \lambda < 1$. Finally, we sample $x_0^\lambda \sim p_\theta(x_0^\lambda|x_t^\lambda)$. The audio samples are in Section VI on the demo website.

## 6 CONCLUSION

In this paper, we present DiffWave, a versatile generative model for raw waveform. In the neural vocoding task, it readily models the fine details of waveform conditioned on mel spectrogram and matches the strong autoregressive neural vocoder in terms of speech quality. In unconditional and class-conditional generation tasks, it properly captures the large variations within the data and produces realistic voices and consistent word-level pronunciations. To the best of our knowledge, DiffWave is the first waveform model that exhibits such versatility. DiffWave raises a number of open problems and provides broad opportunities for future research. For example, it would be meaningful to push the model to generate longer utterances, as DiffWave potentially has very large receptive fields. Second, optimizing the inference speed would be beneficial for applying the model in production TTS, because DiffWave is still slower than flow-based models. We found the most effective denoising steps in the reverse process occur near $x_0$, which suggests an even smaller $T$ is possible in DiffWave. In addition, the model parameters $\theta$ are shared across the reverse process, so the persistent kernels that stash the parameters on-chip would largely speed-up inference on GPUs (Diamos et al., 2016).

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

## A  PROOF OF **PROPOSITION** 1

*Proof.* We expand the ELBO in Eq. (3) into the sum of a sequence of tractable KL divergences below.

$$\text{ELBO} = \mathbb{E}_q \log \frac{p_\theta(x_0, \cdots, x_{T-1}|x_T) \times p_{\text{latent}}(x_T)}{q(x_1, \cdots, x_T|x_0)}$$

$$= \mathbb{E}_q \left( \log p_{\text{latent}}(x_T) - \sum_{t=1}^{T} \log \frac{p_\theta(x_{t-1}|x_t)}{q(x_t|x_{t-1})} \right)$$

$$= \mathbb{E}_q \left( \log p_{\text{latent}}(x_T) - \log \frac{p_\theta(x_0|x_1)}{q(x_1|x_0)} - \sum_{t=2}^{T} \left( \log \frac{p_\theta(x_{t-1}|x_t)}{q(x_{t-1}|x_t, x_0)} + \log \frac{q(x_{t-1}|x_0)}{q(x_t|x_0)} \right) \right)$$

$$= \mathbb{E}_q \left( \log \frac{p_{\text{latent}}(x_T)}{q(x_T|x_0)} - \log p_\theta(x_0|x_1) - \sum_{t=2}^{T} \log \frac{p_\theta(x_{t-1}|x_t)}{q(x_{t-1}|x_t, x_0)} \right)$$

$$= -\mathbb{E}_q \left( \text{KL}\left(q(x_T|x_0)\|p_{\text{latent}}(x_T)\right) + \sum_{t=2}^{T} \text{KL}\left(q(x_{t-1}|x_t, x_0)\|p_\theta(x_{t-1}|x_t)\right) - \log p_\theta(x_0|x_1) \right)$$

$$(9)$$

Before we calculate these terms individually, we first derive $q(x_t|x_0)$ and $q(x_{t-1}|x_t, x_0)$. Let $\epsilon_i$'s be independent standard Gaussian random variables. Then, by definition of $q$ and using the notations of constants introduced in Eq. (4), we have

$$\begin{aligned} x_t &= \sqrt{\alpha_t} x_{t-1} + \sqrt{\beta_t} \epsilon_t \\ &= \sqrt{\alpha_t \alpha_{t-1}} x_{t-2} + \sqrt{\alpha_t \beta_{t-1}} \epsilon_{t-1} + \sqrt{\beta_t} \epsilon_t \\ &= \sqrt{\alpha_t \alpha_{t-1} \alpha_{t-1}} x_{t-3} + \sqrt{\alpha_t \alpha_{t-1} \beta_{t-2}} \epsilon_{t-2} + \sqrt{\alpha_t \beta_{t-1}} \epsilon_{t-1} + \sqrt{\beta_t} \epsilon_t \\ &= \cdots \\ &= \sqrt{\bar{\alpha}_t} x_0 + \sqrt{\alpha_t \alpha_{t-1} \cdots \alpha_2 \beta_1} \epsilon_1 + \cdots + \sqrt{\alpha_t \beta_{t-1}} \epsilon_{t-1} + \sqrt{\beta_t} \epsilon_t \end{aligned}$$

Note that $q(x_t|x_0)$ is still Gaussian, and the mean of $x_t$ is $\sqrt{\bar{\alpha}_t} x_0$, and the variance matrix is $(\alpha_t \alpha_{t-1} \cdots \alpha_2 \beta_1 + \cdots + \alpha_t \beta_{t-1} + \beta_t)I = (1 - \bar{\alpha}_t)I$. Therefore,

$$q(x_t|x_0) = \mathcal{N}(x_t; \ \sqrt{\bar{\alpha}_t} x_0, \ (1 - \bar{\alpha}_t)I). \tag{10}$$

It is worth mentioning that,

$$q(x_T|x_0) = \mathcal{N}(x_T; \ \sqrt{\bar{\alpha}_T} x_0, \ (1 - \bar{\alpha}_T)I), \tag{11}$$

where $\bar{\alpha}_T = \prod_{t=1}^{T}(1 - \beta_t)$ approaches zero with large $T$.

Next, by Bayes rule and Markov chain property,

$$\begin{aligned} q(x_{t-1}|x_t, x_0) &= \frac{q(x_t|x_{t-1}) \, q(x_{t-1}|x_0)}{q(x_t|x_0)} \\ &= \frac{\mathcal{N}(x_t; \sqrt{\alpha_t} x_{t-1}, \beta_t I) \, \mathcal{N}(x_{t-1}; \sqrt{\bar{\alpha}_{t-1}} x_0, (1 - \bar{\alpha}_{t-1})I)}{\mathcal{N}(x_t; \sqrt{\bar{\alpha}_t} x_0, (1 - \bar{\alpha}_t)I)} \\ &= (2\pi\beta_t)^{-\frac{d}{2}} (2\pi(1 - \bar{\alpha}_{t-1}))^{-\frac{d}{2}} (2\pi(1 - \bar{\alpha}_t))^{\frac{d}{2}} \times \\ &\quad \exp\left( -\frac{\|x_t - \sqrt{\alpha_t} x_{t-1}\|^2}{2\beta_t} - \frac{\|x_{t-1} - \sqrt{\bar{\alpha}_{t-1}} x_0\|^2}{2(1 - \bar{\alpha}_{t-1})} + \frac{\|x_t - \sqrt{\bar{\alpha}_t} x_0\|^2}{2(1 - \bar{\alpha}_t)} \right) \\ &= (2\pi\tilde{\beta}_t)^{-\frac{d}{2}} \exp\left( -\frac{1}{2\tilde{\beta}_t} \left\| x_{t-1} - \frac{\sqrt{\bar{\alpha}_{t-1}}\beta_t}{1 - \bar{\alpha}_t} x_0 - \frac{\sqrt{\alpha_t}(1 - \bar{\alpha}_{t-1})}{1 - \bar{\alpha}_t} x_t \right\|^2 \right) \end{aligned}$$

Therefore,

$$q(x_{t-1}|x_t, x_0) = \mathcal{N}(x_{t-1}; \ \frac{\sqrt{\bar{\alpha}_{t-1}}\beta_t}{1 - \bar{\alpha}_t} x_0 + \frac{\sqrt{\alpha_t}(1 - \bar{\alpha}_{t-1})}{1 - \bar{\alpha}_t} x_t, \tilde{\beta}_t I). \tag{12}$$

Now, we calculate each term of the ELBO expansion in Eq. (9). The first constant term is

$$\begin{aligned} \mathbb{E}_q \text{KL}\left(q(x_T|x_0)\|p_{\text{latent}}(x_T)\right) &= \mathbb{E}_{x_0} \text{KL}\left(\mathcal{N}(\sqrt{\bar{\alpha}_T} x_0, (1 - \bar{\alpha}_T)I)\|\mathcal{N}(0, I)\right) \\ &= \frac{1}{2}\mathbb{E}_{x_0} \|\sqrt{\bar{\alpha}_T} x_0 - 0\|^2 + d\left( \log \frac{1}{\sqrt{1 - \bar{\alpha}_T}} + \frac{1 - \bar{\alpha}_T - 1}{2} \right) \\ &= \frac{\bar{\alpha}_T}{2}\mathbb{E}_{x_0} \|x_0\|^2 - \frac{d}{2}(\bar{\alpha}_T + \log(1 - \bar{\alpha}_T)) \end{aligned}$$

Next, we compute $\mathrm{KL}\left(q(x_{t-1}|x_t, x_0)\|p_\theta(x_{t-1}|x_t)\right)$. Because both $q(x_{t-1}|x_t, x_0)$ and $p_\theta(x_{t-1}|x_t)$ are Gaussian with the same covariance matrix $\tilde{\beta}_t I$, the KL divergence between them is $\frac{1}{2\tilde{\beta}_t}$ times the squared $\ell_2$ distance between their means. By the expression of $q(x_t|x_0)$, we have $x_t = \sqrt{\bar{\alpha}_t}x_0 + \sqrt{1 - \bar{\alpha}_t}\epsilon$. Therefore, we have

$$
\begin{aligned}
&\mathbb{E}_q\,\mathrm{KL}\left(q(x_{t-1}|x_t, x_0)\|p_\theta(x_{t-1}|x_t)\right) \\
&= \frac{1}{2\tilde{\beta}_t}\mathbb{E}_{x_0}\left\|\frac{\sqrt{\bar{\alpha}_{t-1}}\beta_t}{1 - \bar{\alpha}_t}x_0 + \frac{\sqrt{\alpha_t}(1 - \bar{\alpha}_{t-1})}{1 - \bar{\alpha}_t}x_t - \frac{1}{\sqrt{\alpha_t}}\left(x_t - \frac{\beta_t}{\sqrt{1 - \bar{\alpha}_t}}\epsilon_\theta(x_t, t)\right)\right\|^2 \\
&= \frac{1}{2\tilde{\beta}_t}\mathbb{E}_{x_0,\epsilon}\left\|\frac{\sqrt{\bar{\alpha}_{t-1}}\beta_t}{1 - \bar{\alpha}_t}\cdot\frac{x_t - \sqrt{1 - \bar{\alpha}_t}\epsilon}{\sqrt{\bar{\alpha}_t}} + \frac{\sqrt{\alpha_t}(1 - \bar{\alpha}_{t-1})}{1 - \bar{\alpha}_t}x_t - \frac{1}{\sqrt{\alpha_t}}\left(x_t - \frac{\beta_t}{\sqrt{1 - \bar{\alpha}_t}}\epsilon_\theta(x_t, t)\right)\right\|^2 \\
&= \frac{1}{2\tilde{\beta}_t}\cdot\frac{\beta_t^2}{\alpha_t(1 - \bar{\alpha}_t)}\mathbb{E}_{x_0,\epsilon}\left\|0\cdot x_t + \epsilon - \epsilon_\theta(x_t, t)\right\|^2 \\
&= \frac{\beta_t^2}{2\frac{1 - \bar{\alpha}_{t-1}}{1 - \bar{\alpha}_t}\beta_t\alpha_t(1 - \bar{\alpha}_t)}\mathbb{E}_{x_0,\epsilon}\left\|\epsilon - \epsilon_\theta(x_t, t)\right\|^2 \\
&= \frac{\beta_t}{2\alpha_t(1 - \bar{\alpha}_{t-1})}\mathbb{E}_{x_0,\epsilon}\left\|\epsilon - \epsilon_\theta(x_t, t)\right\|^2
\end{aligned}
$$

Finally, as $x_1 = \sqrt{\bar{\alpha}_1}x_0 + \sqrt{1 - \bar{\alpha}_1}\epsilon = \sqrt{\alpha_1}x_0 + \sqrt{1 - \alpha_1}\epsilon$, we have

$$
\begin{aligned}
\mathbb{E}_q\log p_\theta(x_0|x_1) &= \mathbb{E}_q\log\mathcal{N}\left(x_0; \frac{1}{\sqrt{\alpha_1}}\left(x_1 - \frac{\beta_1}{\sqrt{1 - \alpha_1}}\epsilon_\theta(x_1, 1)\right), \beta_1 I\right) \\
&= \mathbb{E}_q\left(-\frac{d}{2}\log 2\pi\beta_1 - \frac{1}{2\beta_1}\left\|x_0 - \frac{1}{\sqrt{\alpha_1}}\left(x_1 - \frac{\beta_1}{\sqrt{1 - \alpha_1}}\epsilon_\theta(x_1, 1)\right)\right\|^2\right) \\
&= -\frac{d}{2}\log 2\pi\beta_1 - \frac{1}{2\beta_1}\mathbb{E}_{x_0,\epsilon}\left\|x_0 - \frac{1}{\sqrt{\alpha_1}}\left(\sqrt{\alpha_1}x_0 + \sqrt{1 - \alpha_1}\epsilon - \frac{\beta_1}{\sqrt{1 - \alpha_1}}\epsilon_\theta(x_1, 1)\right)\right\|^2 \\
&= -\frac{d}{2}\log 2\pi\beta_1 - \frac{1}{2\beta_1}\mathbb{E}_{x_0,\epsilon}\left\|\frac{\sqrt{\beta_1}}{\sqrt{\alpha_1}}(\epsilon - \epsilon_\theta(x_1, 1))\right\|^2 \\
&= -\frac{d}{2}\log 2\pi\beta_1 - \frac{1}{2\alpha_1}\mathbb{E}_{x_0,\epsilon}\|\epsilon - \epsilon_\theta(x_1, 1)\|^2
\end{aligned}
$$

The computation of the ELBO is now finished. $\qquad\square$

## B  DETAILS OF THE FAST SAMPLING ALGORITHM

Let $T_{\text{infer}} \ll T$ be the number of steps in the reverse process (sampling) and $\{\eta_t\}_{t=1}^{T_{\text{infer}}}$ be the user-defined variance schedule, which can be independent with the training variance schedule $\{\beta_t\}_{t=1}^T$. Then, we compute the corresponding constants in the same way as Eq. (4):

$$\gamma_t = 1 - \eta_t, \;\; \bar{\gamma}_t = \prod_{s=1}^t \gamma_s, \;\; \tilde{\eta}_t = \frac{1 - \bar{\gamma}_{t-1}}{1 - \bar{\gamma}_t} \eta_t \;\; \text{for} \;\; t > 1 \;\; \text{and} \;\; \tilde{\eta}_1 = \eta_1. \tag{13}$$

As step $s$ during sampling, we need to select an $t$ and use $\epsilon_\theta(\cdot, t)$ to eliminate noise. This is realized by aligning the noise levels from the user-defined and the training variance schedules. Ideally, we want $\sqrt{\bar{\alpha}_t} = \sqrt{\bar{\gamma}_s}$. However, since this is not always possible, we interpolate $\sqrt{\bar{\gamma}_s}$ between two consecutive training noise levels $\sqrt{\bar{\alpha}_{t+1}}$ and $\sqrt{\bar{\alpha}_t}$, if $\sqrt{\bar{\gamma}_s}$ is between them. We therefore obtain the desired aligned diffusion step $t$, which we denote $t_s^{\text{align}}$, via the following equation:

$$t_s^{\text{align}} = t + \frac{\sqrt{\bar{\alpha}_t} - \sqrt{\bar{\gamma}_s}}{\sqrt{\bar{\alpha}_t} - \sqrt{\bar{\alpha}_{t+1}}} \;\;\; \text{if} \; \sqrt{\bar{\gamma}_s} \in [\, \sqrt{\bar{\alpha}_{t+1}}, \sqrt{\bar{\alpha}_t} \,]. \tag{14}$$

Note that, $t_s^{\text{align}}$ is floating-point number, which is different from the integer diffusion-step at training.

Finally, the parameterizations of $\mu_\theta$ and $\sigma_\theta$ are defined in a similar way as Eq. (5):

$$\mu_\theta^{\text{fast}}(x_s, s) = \frac{1}{\sqrt{\gamma_s}}\left( x_s - \frac{\eta_s}{\sqrt{1 - \bar{\gamma}_s}}\epsilon_\theta(x_s, t_s^{\text{align}}) \right), \;\; \text{and} \;\; \sigma_\theta^{\text{fast}}(x_s, s) = \tilde{\eta}_s^{\frac{1}{2}}. \tag{15}$$

The fast sampling algorithm is summarized in Algorithm 3.

---

**Algorithm 3** Fast Sampling

Sample $x_{T_{\text{infer}}} \sim p_{\text{latent}} = \mathcal{N}(0, I)$
**for** $s = T_{\text{infer}}, T_{\text{infer}} - 1, \cdots, 1$ **do**
    Compute $\mu_\theta^{\text{fast}}(x_s, s)$ and $\sigma_\theta^{\text{fast}}(x_s, s)$ using Eq. (15)
    Sample $x_{s-1} \sim \mathcal{N}(x_{s-1}; \mu_\theta^{\text{fast}}(x_s, s), \sigma_\theta^{\text{fast}}(x_s, s)^2 I)$
**end for**
**return** $x_0$

---

In neural vocoding task, we use user-defined variance schedules $\{0.0001, 0.001, 0.01, 0.05, 0.2, 0.7\}$ for DiffWave $_{\text{LARGE}}$ and $\{0.0001, 0.001, 0.01, 0.05, 0.2, 0.5\}$ for DiffWave $_{\text{BASE}}$ in Section 5.1.

The fast sampling algorithm is similar to the sampling algorithm in Chen et al. (2020) in the sense of considering the noise levels as a controllable variable during sampling. However, the fast sampling algorithm for DiffWave does not need to modify the training procedure (Algorithm 1), and can just reuse the trained model checkpoint with large $T$.

## C    DETAILS OF THE MODEL ARCHITECTURE

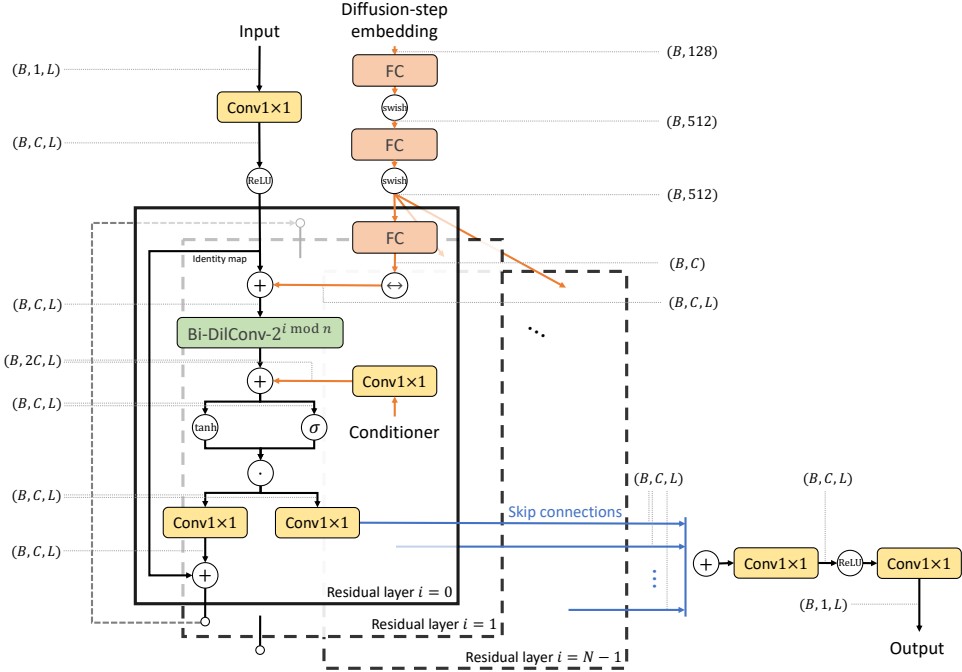

Figure 3: The network architecture of DiffWave in modeling $\epsilon_\theta(x_t, t)$, including tensor shapes at each stage and activation functions. $B$ is the batch size, $C$ is the number of residual/skip channels of the network, and $L$ is data dimension.

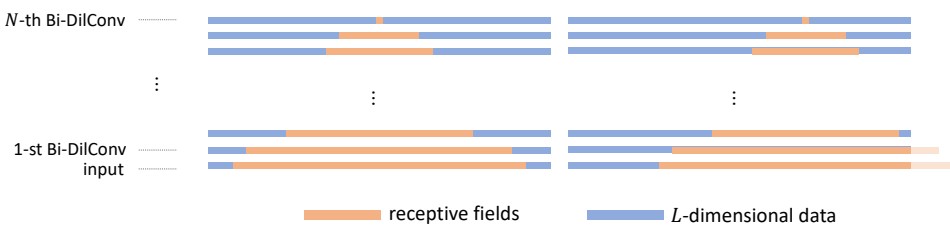

Figure 4: The Receptive fields of the output units within DiffWave network.

# D DETAILS OF AUTOMATIC EVALUATION METRICS IN SECTION 5.2 AND 5.3

The automatic evaluation metrics used in Section 5.2 and 5.3 are described as follows. Given an input audio $x$, an 1024-dimensional feature vector (denoted as $\mathcal{F}_{\text{feature}}(x)$) is computed by the ResNeXT $\mathcal{F}$, and is then transformed to the 10-dimensional multinomial distribution (denoted as $p_{\mathcal{F}}(x)$) with a fully connected layer and a softmax layer. Let $X_{\text{train}}$ be the trainset, $p_{\text{gen}}$ be the distribution of generated data, and $X_{\text{gen}} \sim p_{\text{gen}}(i.i.d.)$ be the set of generated audio samples. Then, we compute the following automatic evaluation metrics:

- **Fréchet Inception Distance (FID)** (Heusel et al., 2017) computes the Wasserstein-2 distance between Gaussians fitted to $\mathcal{F}_{\text{feature}}(X_{\text{train}})$ and $\mathcal{F}_{\text{feature}}(X_{\text{gen}})$. That is,

$$\text{FID} = \|\mu_g - \mu_t\|^2 + \text{Tr}\left(\Sigma_t + \Sigma_g - 2(\Sigma_t \Sigma_g)^{\frac{1}{2}}\right),$$

  where $\mu_t, \Sigma_t$ are the mean vector and covariance matrix of $\mathcal{F}_{\text{feature}}(X_{\text{train}})$, and where $\mu_g, \Sigma_g$ are the mean vector and covariance matrix of $\mathcal{F}_{\text{feature}}(X_{\text{gen}})$.

- **Inception Score (IS)** (Salimans et al., 2016) computes the following:

$$\text{IS} = \exp\left(\mathbb{E}_{x \sim p_{\text{gen}}} \text{KL}\left(p_{\mathcal{F}}(x) \| \mathbb{E}_{x' \sim p_{\text{gen}}} p_{\mathcal{F}}(x')\right)\right),$$

  where $\mathbb{E}_{x' \sim p_{\text{gen}}} p_{\mathcal{F}}(x')$ is the marginal label distribution.

- **Modified Inception Score (mIS)** (Gurumurthy et al., 2017) computes the following:

$$\text{mIS} = \exp\left(\mathbb{E}_{x, x' \sim p_{\text{gen}}} \text{KL}\left(p_{\mathcal{F}}(x) \| p_{\mathcal{F}}(x')\right)\right).$$

- **AM Score** (Zhou et al., 2017) computes the following:

$$\text{AM} = \text{KL}\left(\mathbb{E}_{x' \sim q_{\text{data}}} p_{\mathcal{F}}(x') \| \mathbb{E}_{x \sim p_{\text{gen}}} p_{\mathcal{F}}(x)\right) + \mathbb{E}_{x \sim p_{\text{gen}}} \text{H}(p_{\mathcal{F}}(x)),$$

  where $\text{H}(\cdot)$ computes the entropy. Compared to IS, AM score takes into consideration the the prior distribution of $p_{\mathcal{F}}(X_{\text{train}})$.

- **Number of Statistically-Different Bins (NDB)** (Richardson & Weiss, 2018): First, $X_{\text{train}}$ is clustered into $K$ bins by $K$-Means in the feature space (where $K = 50$ in our evaluation). Next, each sample in $X_{\text{gen}}$ is assigned to its nearest bin. Then, NDB is the number of bins that contain statistically different proportion of samples between training samples and generated samples.

