# OpenReview forum: "DiffWave: A Versatile Diffusion Model for Audio Synthesis"
_ICLR.cc/2021/Conference — ICLR 2021 Oral_

### Official Review · AnonReviewer3 · 2020-10-28
**Paper showing the applicability of Diffusion Probabilistic model for speech synthesis tasks**

**Rating:** 7
**Confidence:** 5

**Review:**

The Diffusion Probabilistic model is gaining popularity as a generative model. Diffwave explores the same for speech synthesis tasks. They show very good results i.e. matching autoregressive Wavenet on the conditional and outperforming baselines on the unconditional audio waveform synthesis tasks.

The paper is clearly written, and easy to follow. The work does not provide any novel machine learning or generative modeling insights. However, the work is significant for speech synthesis applications since it shows great results, with a small foot-print network with a very new method. This work can be expected to spark a plethora of follow-up works for speech synthesis and other real-valued time-series modeling tasks.

Pros:
1. Very good results on conditional neural vocoding task.
2. State-of-the-art results on unconditional speech synthesis task

Cons:
1. Lack of novelty in terms of insights/approach
2. Motivation and ablation-study for various design choices are missing

*Further Comments*: An ablation study that establishes the impact of various hyper-parameters/components (e.g. choice of diffusion step embedding function, more detailed analysis of width/depth of the network, etc.) would help the readers get a lot more value out of the paper. A qualitative study of samples, specifically, pointing out any bias that underlies generative modeling via the diffusion process would be great.

---

> ### Author Response · Authors · 2020-11-23
> **Response to Reviewer 3**
>
> Thank you so much for your review. We will address your comments in the following.
>
> “Lack of novelty in terms of insights/approach”
> * This work focuses on real-world speech/audio synthesis application, and provides successful recipes and non-trivial insights for building state-of-the-art generative models for raw audio. In particular, the unconditional generation results demonstrate that diffusion probabilistic models with limited capacity can capture the major variations within the data, and provide very large receptive fields for modeling very long sequences (16,000 time-steps) without any conditional information. In contrast, the state-of-the-art likelihood-based model (i.e., WaveNet) fail to do that, because they are forced to learn all possible modes within the data by minimizing the forward KL divergence between the model and empirical distribution. The state-of-the-art GAN-based models (i.e., WaveGAN, Parallel WaveGAN) also produce inferior results , due to the noticeable challenge of modeling very long raw waveform unconditionally. We think such success and insights for an important domain (speech/audio) are as valuable as novel methodology development.
>
> “Motivation and ablation-study for various design choices are missing. Further Comments: An ablation study that establishes the impact of various hyper-parameters/components (e.g. choice of diffusion step embedding function, more detailed analysis of width/depth of the network, etc.) would help the readers get a lot more value out of the paper. ”
> * It’s really expensive to perform systematic ablation-study (e.g., various hyper-parameters), mainly because the current quantitative evaluation for speech synthesis relies on costly human evaluation. However, we do have a lot of qualitative observations when we tune the model.  We are happy to discuss these observations and results under different hyperparameters in the camera ready version, including the number of diffusion steps, residual channels, fully-connected layer widths, etc.
>
> “A qualitative study of samples, specifically, pointing out any bias that underlies generative modeling via the diffusion process would be great.”
> * Many thanks for your comment. Indeed, we find the audio samples from DiffWave vocoder tend to be “clearer” than likelihood models (WaveNet, WaveFlow), while the samples from likelihood models are more similar to the ground-truth audio samples. We think the underlying reason for such bias is that likelihood models are forced to model all the details within ground-truth data, including some artifacts from the recording environment, which can make synthesized audio more realistic. In contrast, the denoising diffusion model optimizes a variant of ELBO, thus tends to capture the major variations (i.e. speech part) with limited model capacity and can generate clearer samples.

---

### Official Review · AnonReviewer2 · 2020-10-29
**Well-written paper with strong, well-presented results.**

**Rating:** 8
**Confidence:** 3

**Review:**

Well-written paper with strong, well-presented results. The MOS results attain or surpass the best WaveNet results. The presentation is on the whole clear.

I was a bit confused by the core model description at first. In particular, the index t on the samples x_t is not a time-index, correct? Rather, it's just a step in the diffusion process? At first I thought it was a time-index, and so the model seemed very much like an AR model, leaving me very confused.

Some additional things I liked about the work:

Compelling contrast with existing methods (WaveNet, VAE, GANs).
Use of multiple metrics in the evaluation, 5 objective metrics in addition to MOS, e.g. Tables 2 & 3, with details on the metrics provided in an Appendix.
Use of multiple models as reference models, WaveNet, WaveGlow, WaveFlow, Clarinet, WaveGAN, in addition to the proposed model.
Focus on unconditional generation, which AIU has not received that much attention in the community
Where I am unsure is the originality of the work. I personally am not aware of the diffusion approach having been applied to TTS, but this is not my primary area of expertise. Obviously, if there is related work in TTS with diffusion models, this should be cited.

Also, what I don't see in the Conclusion is any discussion of the weaknesses and challenges for the model going forward. The paper would be strengthened by having a more balanced conclusion.

Some caveats regarding my review:

I am not familiar with the specific datasets used, so cannot fully appreciate the significance of the results reported.
I did not check the math in detail; the notation overall seemed clear and consistent to me (though see my first set of comments).

I have a few more specific comments.

Throughout the paper, "... audios ..." : "audio" is not usually used as a plural noun.

E.g. "We randomly generate 1,000 audios" --> "We randomly generate 1,000 audio waveforms"?

"Notably, the quality of audios ... " --> "Notably, the quality of audio ... "

"Note that, the quality of ground-truth ...": nit, no comma after "that".

---

> ### Author Response · Authors · 2020-11-23
> **Response to Reviewer 2**
>
> Thank you so much for your detailed review. They are very helpful to improve the quality of our paper.
>
>  ”I was a bit confused by the core model description at first. In particular, the index t on the samples x_t is not a time-index, correct? Rather, it's just a step in the diffusion process?“
> * Yes, the subscription t refers to the diffusion step. Sorry for the confusion. It’s our fault to denote x_i (typo: should be x_t) at the beginning of Section 2, but use x_t in the other part of the paper. We have revised the paper accordingly.
>
> ”Also, what I don't see in the Conclusion is any discussion of the weaknesses and challenges for the model going forward. The paper would be strengthened by having a more balanced conclusion.“
> * We have extended the conclusion section with discussion of weaknesses, challenges and future directions in the updated draft.
>
> “Audios -> Audio” & “Note that,  -> Note that”
> * We appreciate your specific comments. We have addressed all your comments in the updated version.

---

### Official Review · AnonReviewer1 · 2020-10-30
**State of the art neural vocoder that is fast, small, accurate, and works with little supervision.**

**Rating:** 9
**Confidence:** 4

**Review:**

This paper describes a neural vocoder based on a diffusion probabilistic model. The model utilizes a fixed-length markov chain to convert between a latent uncorrelated Gaussian vector and a full-length observation. The conversion from observation to latent is fixed and amounts to adding noise at each step. The conversion from latent to observation reveals slightly more of the observation from the latent at each step via a sort of cancellation. This process is derived theoretically based on maximizing the variational lower bound (ELBO) of the model and follows Ho et al. (2020) who derived it for image generation. Thorough experiments show that the model produces high quality speech syntheses on the LJ dataset (MOS comparable to WaveNet and real speech) when conditionally synthesizing from the true mel spectrogram, while generating much more quickly than WaveNet. Perhaps more interesting and surprising, however, is that it generates very high quality and intelligible short utterances with no conditioning, and also admits to global conditioning, e.g., with a digit label.

The paper is very clearly written, with the description of the model going into sufficient detail in the main body of the paper for the reader to understand it without getting bogged down in the less immediately relevant details. The experiments are thorough and well executed, comparing with listening tests to many state of the art neural vocoders for the conditional task. It describes a thorough evaluation of the unconditional generation task, which is in general difficult, but in this case was constrained in such a way as to make it feasible and informative, using reasonable metrics that clearly show the advantages of the proposed approach. The literature review is thorough and comes at a point in the paper where the reader understands the proposed approach and can appreciate the nuances of the differences between the approaches. Audio samples are provided on a companion website and demonstrate the effectiveness of the approach along with some additional interesting properties of the model not even mentioned in the paper (denoising most impressively, interpolation between speakers is less convincing).

The paper has two minor weaknesses. First is that it does not make a clearer distinction between the concurrent work from Chen et al (2020) along similar lines, although presumably this paper was not released prior to submission of the current paper. An extended comparison would be welcome in a camera ready version of this paper. Second, that it doesn't explicitly state the real-time factor of WaveNet generation in the results discussion on page 6, which is presumably much smaller than 1. This section compares to WaveNet in terms of quality and WaveFlow in terms of speed, slightly being slightly worse in both comparisons, but better in the opposite, partially missing, comparisons.

Overall, this paper makes a strong contribution to the field of neural vocoding and to the field of representation learning more generally for long-duration intricately structured signals (i.e., speech).

---

> ### Author Response · Authors · 2020-11-23
> **Response to Reviewer 1**
>
> Many thanks for your detailed review & suggestion; they are really helpful to improve the quality of our paper.
>
> “First is that it does not make a clearer distinction between the concurrent work from Chen et al (2020) along similar lines, although presumably this paper was not released prior to submission of the current paper. An extended comparison would be welcome in a camera ready version of this paper.”
> * Thanks for the suggestion. We will extend the comparison in the final version of this paper. There are several differences between our work and Chen et al (2020):
> - i) Task: Chen et al (2020) focuses on neural vocoding tasks (e.g., conditioned on mel spectrogram). In contrast, we investigate broader waveform generation with various levels of conditional information. We think the unconditional generation results are particularly interesting, given the previous inferior results in raw audio domain vs. the impressive results in image domain.
> - ii) Architecture: We use a WaveNet based architecture, which is compact and flexible for various waveform generation tasks. In contrast, the GAN-TTS based architecture in Chen et al (2020) is tailored for neural vocoding, and it would be non-trivial to adapt for other tasks.
> - iii) Model footprint: For neural vocoding, the model size in Chen et al (2020) ranges from 15M (base model) to 23M (large model) parameters, while our DiffWave has 2.64M parameters (residual channels = 64) and 6.91M parameters (residual channels = 128), respectively.
> - iv) Training cost: DiffWave requires a much smaller batch size (16 vs. 256) and fewer computational resources at training. The models in Chen et al (2020) are trained on 32 TPU v2 cores (base model) and 128 TPU v3 cores (large model), respectively. DiffWave is trained on 8 NVIDIA 2080Ti GPUs.
>
>
> “Second, that it doesn't explicitly state the real-time factor of WaveNet generation in the results discussion on page 6, which is presumably much smaller than 1.”
> * Thanks for the reminder. We use an implementation of WaveNet without engineered kernels, which is 500 times slower than real time for 22.05 kHz audio. We have stated it in our revision accordingly.

---

### Official Review · AnonReviewer4 · 2020-11-03
**An interesting paper showing good results on applying denoising diffusion processes to spectrogram inversion.**

**Rating:** 7
**Confidence:** 5

**Review:**

The paper develops a speech synthesis model using denoising diffusion processes, a generative model framework recently demonstrated in image generation (Ho et al. 2020).    The application is straightforward and there is little if any theoretical difference from the Ho et al. paper.    I didn't check the proofs included in the appendix, but they along with the learning and sampling procedure seem to be already developed in (Ho et al. 2020).  The authors should take care to be very clear about the mathematical developments that are directly taken from the prior literature, and what developments are introduced in this paper.
Nevertheless the experiments on this application is valuable, and makes significant progress on a problem that has proven surprisingly difficult to solve in an efficient way.
The experiments and demos are convincing, and the results could be considered highly competitive in conditional generation and  state of the art for class-conditional and non-conditional generation.

The writing at times could use improvement.   In the abstract, line 1, "we propose DiffWave, a versatile Diffusion probabilistic model for conditional and unconditional Waveform generation".   I don't like this style of capitalizing things in a sentence that are not proper nouns.  If you want to introduce an abbreviation derived from a term or phrase, a widely accepted conventional method is to italicize the phrase and define the acronym the first time it is used, as in "\emph{diffusion waveform} (DiffWave) model".   Later you have "DiffWave produces high-fidelity audios in Different Waveform generation tasks":  Why is "Different Waveform" capitalized?   DiffWave has already been defined relative to "diffusion waveform".   If this is supposed to be cute, it's not.
Defining the acronym in the abstract is OK, but not necessary.  In any case, you still have to define it again in the body of the paper, since the abstract is considered a standalone summary of your document.
Also "audios" is not a word.  Please use "audio signals". This is repeated throughout the paper.
Other examples:
"This avoids the ... issues *stemmed from the joint training"   stemmed -->  stemming
" for generating very long waveform" :  waveform --> waveforms

---

> ### Author Response · Authors · 2020-11-23
> **Response to Reviewer 4**
>
> Thank you for the detailed comments. They are really helpful for improving our paper.
>
>
> “The authors should take care to be very clear about the mathematical developments that are directly taken from the prior literature, and what developments are introduced in this paper.”
> * Thanks for the comment. We have updated the draft to make it clearer.
>
>
> “The writing at times could use improvement...”
> * We appreciate your suggestions regarding writing. We have addressed all your comments in the updated version.

---

### Official Review · AnonReviewer5 · 2020-11-06
**Review of DiffWave**

**Rating:** 7
**Confidence:** 3

**Review:**

Summary:

The authors adapt the recent trend of work on denoising diffusion probablistic models to the task of conditional or unconditional waveform generation.
Using the same principles as in (Ho et al 2020), as well as a Wavenet-like non causal model, the authors provide state of the art results for both tasks, as evaluated on spoken digits dataset (for unconditional and conditional generation) and on the LJ speech dataset (for deep vocoding).
The model proposed is faster to evaluate than WaveNet, and has less parameters than WaveGlow. It achieves a slightly higher MOS than WaveFlow for a comparable model size. The generation speed is comparable to previous methods.

## Review

The paper is clear, well structured and the authors provide many experiments to validate their approach.
While serious, the paper does lack novelty, as the method is completely taken from (Ho et al. 2020). The architecture is similar to Wavenet, but non causal. Note that this exact same non causal wavenet architecture has already been used for source separation [Rethage et al. 2018, Lluis et al. 2019].

One limitation is that unconditional, or weakly conditioned generation (i.e. not conditioned on a mel-spectrogram) is only evaluated on single digit generation, which is relatively limite. While the samples shows an improvement over WaveNet, it seems the proposed architecture would still struggle to generate longer sequences, like an entire sentence for instance.
It would be interesting to add WaveGlow or WaveFlow to the SC09 comparison.

Overall I recommend acceptance as the paper show that denoising diffusion process can be used for waveform generation, even though the paper does not bring further novelty.

References:
Rethage et al. 2018: A Wavenet for Speech Denoising
Lluis et al. 2019: End-to-end music source separation: is it possible in the waveform domain?

---

> ### Author Response · Authors · 2020-11-23
> **Response to Reviewer 5**
>
> Thank you so much for your review. We will address your comments in the following.
>
>
> “Note that this exact same non causal wavenet architecture has already been used for source separation [Rethage et al. 2018, Lluis et al. 2019].”
> - Thanks for pointing them out. We have referenced both papers in our revision.
>
>
> “One limitation is that unconditional, or weakly conditioned generation (i.e. not conditioned on a mel-spectrogram) is only evaluated on single digit generation, which is relatively limited. While the samples show an improvement over WaveNet, it seems the proposed architecture would still struggle to generate longer sequences, like an entire sentence for instance.  It would be interesting to add WaveGlow or WaveFlow to the SC09 comparison.”
> - It is very challenging to directly generate sentence-level utterance in the waveform domain in an unconditional way. In contrast, compressing the raw waveform into compact latent code (e.g, VQ-VAE), then training a generative model (separately) in the latent domain would be much easier to obtain good results. Still, it would be a very interesting problem to work on; we will leave it for future study. In addition, we will try to include the unconditional generation results of WaveFlow in the camera ready version.

---

> ### Comment · ~Wei_Ping1 · 2021-05-19
> **WaveFlow on SC09 for unconditional generation**
>
> Per your suggestion, I've tried WaveFlow on SC09 for unconditional generation. The result is discouraging. The generated samples are mostly not intelligible. I speculate that the deterministic mapping from latents to audios in flow-based model is not quite suitable for this unconditional generation task, because the waveforms exhibit significant amount of stochasticity without strong conditioner (e.g., mel spectrogram in neural vocoder). In contrast, the stochastic mapping from latents to audios in diffusion model seems to be a good fit here.
>
> Many thanks!

---

### Decision · Program_Chairs · 2021-01-07
**Final Decision**

**Decision:**

Accept (Oral)

**Comment:**

I join all five reviewers in recommending acceptance.

There was some discussion about a comparison with WaveGrad (Chen et al., 2020), a contemporaneous work that explores a similar modelling approach for speech generation. While I agree that such a comparison is a useful addition to the manuscript, I do not think it is reasonable to request anything beyond an acknowledgement and citation of the work from the authors as a condition for acceptance. Further discussion and comparison experiments could be valuable, but I believe that should not factor into the final decision. My position is most similar to Reviewer 4's in this sense. The current version of the manuscript briefly discusses the differences between WaveGrad and DiffWave, which I think is more than sufficient. (As an aside, another difference potentially worth discussing is that the "noise schedule" for WaveGrad can be adapted at inference time, enabling a trade-off between inference speed and sample quality, which I believe is not possible for DiffWave in its current form.)

There was some debate about the weakly conditioned generation results; I believe they are a nice addition to the paper, although it would have been suitable for publication without them. They certainly do not detract from it, and might inspire further work in weakly conditioned audio generation (e.g. music). There were also concerns about the clarity of writing, which I believe the authors have addressed in the current version of the manuscript.

This work stands out because it applies a relatively fresh idea in generative modelling to a domain of great practical importance, which has long been dominated by traditional likelihood-based models, with compelling results. While this implies a limited degree of technical novelty, I do not think that is grounds for rejection, and in fact I would argue that making new ideas work well for practical problems is just as important.

---

> ### Author Response · Authors · 2021-04-05
> **Fast sampling in DiffWave vs. "noise schedule" in WaveGrad**
>
> Many thanks for your comment.
>
> Our DiffWave model, which is trained with  $T = 200$ or $50$ diffusion steps, can generate high quality audio within a few $T_{\text{infer}} = 6$ steps at synthesis. See more details at the end of Section 2 in our paper. Note that, our fast sampling method is almost free. It doesn't need any re-training, and only results in very minor perceptual difference. This is different but related to the "noise schedule" used in WaveGrad.